

# Genetic algorithm-based re-optimization of the Schrock catalyst for dinitrogen fixation

Magnus Strandgaard[1], Julius Seumer[1], Bardi Benediktsson[2], Arghya Bhowmik[2], Tejs Vegge[2] and Jan H. Jensen[1]

[1] Department of Chemistry, University of Copenhagen, Copenhagen, Denmark
[2] DTU Energy, Technical University of Denmark, Kgs. Lyngby, Denmark

## ABSTRACT

This study leverages a graph-based genetic algorithm (GB-GA) for the design of efficient nitrogen-fixing catalysts as alternatives to the Schrock catalyst, with the aim to improve the energetics of key reaction steps. Despite the abundance of nitrogen in the atmosphere, it remains largely inaccessible due to its inert nature. The Schrock catalyst, a molybdenum-based complex, offered a breakthrough but its practical application is limited due to low turnover numbers and energetic bottlenecks. The genetic algorithm in our study explores the chemical space for viable modifications of the Schrock catalyst, evaluating each modified catalyst's fitness based on reaction energies of key catalytic steps and synthetic accessibility. Through a series of selection and optimization processes, we obtained fully converged catalytic cycles for 20 molecules at the B3LYP level of theory. From these results, we identified three promising molecules, each demonstrating unique advantages in different aspects of the catalytic cycle. This study offers valuable insights into the potential of generative models for catalyst design. Our results can help guide future work on catalyst discovery for the challenging nitrogen fixation process.

## INTRODUCTION

A previous version of this article was deposited on a preprint server (*Strandgaard et al., 2023*). Nitrogen fixation, a critical process for sustaining life on Earth, plays an essential role in the global nitrogen cycle, and provides bioavailable nitrogen for the growth and development of all living organisms. Although the atmosphere is composed of approximately 78% nitrogen, its inert nature renders it inaccessible to most life forms. Nature has evolved an intricate mechanism to overcome this barrier, primarily through the activity of nitrogen-fixing microorganisms capable of reducing dinitrogen ($N_2$) into bioavailable forms such as ammonia ($NH_3$). Nitrogen fixation driven by transition metal complexes offers a less energy intensive alternative to the conventional Haber-Bosch process (*Westhead et al., 2023*). These complexes can effectively catalyze the conversion of $N_2$ to $NH_3$ under similar conditions as the nitrogenase enzymes in nature. A key breakthrough was Schrock's discovery of molybdenum-based complexes capable of binding

Corresponding author
Jan H. Jensen, jhjensen@chem.ku.dk

and reducing dinitrogen to ammonia under ambient conditions. His work led to the development of the first well-defined, homogeneous catalyst containing a single metal site for nitrogen fixation, known as the Schrock catalyst ([Mo($^{HIPT}$N$_3$N)]) (*Yandulov & Schrock, 2003*; *Schrock, 2005*; *Schrock, 2008*). This molybdenum-based catalyst operates through a series of proton-coupled electron transfer steps, which reduce dinitrogen to ammonia. The Schrock catalyst represents a significant milestone in the field of small molecule nitrogen fixation, as it was the first well-defined, homogeneous catalyst containing a single metal site capable of converting dinitrogen to ammonia. However, its practical application in large-scale nitrogen fixation has been limited due to several factors, including low turnover numbers (*Yandulov & Schrock, 2003*). The Schrock catalyst is the best studied molecular catalyst for dinitrogen reduction, both computationally and experimentally and studies indicate that the last steps in the catalytic cycle are the energetic bottlenecks due to their almost thermoneutral nature. For example, the equilibrium constant for replacing NH$_3$ with N$_2$ on the catalyst (NH$_3 \rightleftharpoons$ N$_2$) has been experimentally measured to be about 0.1, and the reaction energy of the final reduction step (NH$_3^+ \rightarrow$ NH$_3$) has been measured to be between 0 and 1 kcal/mol (*Schrock, 2008*). These findings have been further corroborated and augmented by DFT calculations by Reiher, Neese, Tuczek and others (*Reiher, Le Guennic & Kirchner, 2005*; *Studt & Tuczek, 2005*; *Schenk, Kirchner & Reiher, 2009*; *Thimm et al., 2015*; *Husch & Reiher, 2017*). For example, Fig. 1 shows the catalytic cycle and the corresponding DFT-free energy profile computed by *Thimm et al. (2015)*. While this energy profile shows a large energy increase upon the addition of the first proton (N$_2 \rightarrow$ N$_2$H$^+$) computational studies by *Schenk et al. (2008)* found a more facile route where the proton first binds to one of the N atoms on the ligands before transferring to the bound dinitrogen. Therefore, both DFT free energy calculations and experiments suggest that the main bottlenecks are the last reduction step NH$_3^+ \rightarrow$ NH$_3$ and/or the displacement reaction of a bound NH$_3$ for N$_2$ (NH$_3 \rightarrow$ N$_2$). Further computation studies by *Schenk, Kirchner & Reiher (2009)* indicate two possible paths for the displacement step. Release of NH$_3$ followed by uptake of N$_2$, or *via* an intermediate state where both NH$_3$ and N$_2$ are bound to the molybdenum atom (NH$_3$–N$_2$).

Genetic algorithms have proven to be an effective tool for chemical space exploration (*Brown et al., 2019*; *Leguy et al., 2020*; *Henault, Rasmussen & Jensen, 2020*; *Jensen, 2019*). A main advantage is that the generation and curation of training data is not needed for such methods as ligands can be evaluated on the fly by quantum-methods. Stochastic crossover combined with quantum-method guided optimization means that GA based methods can be easier to interpret than the notorious black box machine learning based methods. In this study, we apply a genetic algorithm to search for alternatives to the hexa-iso-propyl-terphen (HIPT) substituents that make the catalyst have favourable reaction free energies for the last two catalytic steps. The GA discovered substituents are validated at the DFT level of theory with the TZVP basis set and PBE/B3LYP functionals. Furthermore, the goal is to obtain DFT calculated catalytic cycles for promising GA substituent candidates and from these determine promising substituents. The design of catalysts using generative models is still in its infancy (*Chu et al., 2012*; *Seumer et al., 2023*; *Laplaza, Gallarati & Corminboeuf, 2022*) and requires additional work to fully exploit the methods potential. This is a preliminary

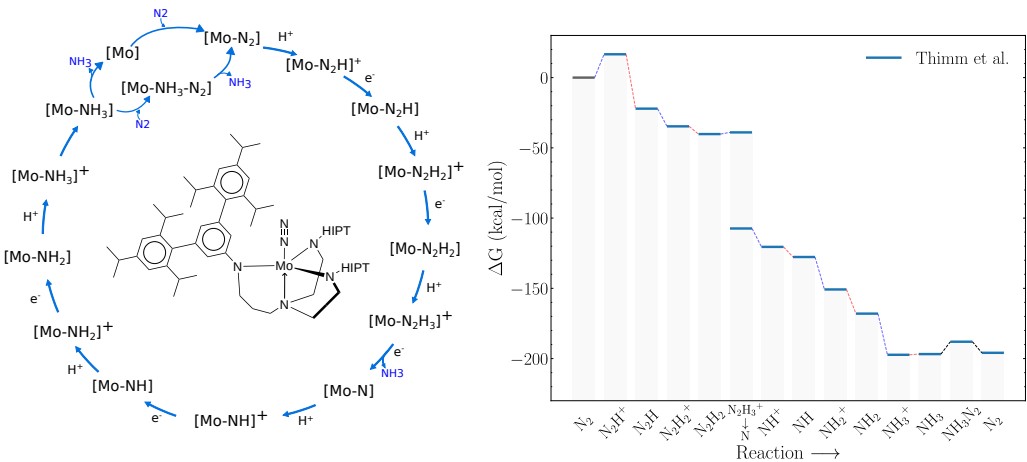

**Figure 1** **Schematic of the Schrock cycle (left) and the free energy profile of the cycle as calculated by** *Thimm et al.* **(right).** The blue lines connecting intermediate states in the energy profile indicate protonation steps, red lines indicate reduction steps and black lines indicate the chemical steps of $N_2$ binding and $NH_3$ release.

study, where we determine the feasibility of this ambitious goal. Thus, we do not include important design considerations such as the steric protection of the the Mo atom to avoid $H^+$ reduction or dimerization of the catalyst.

## COMPUTATIONAL METHODOLOGY

The workflow implemented in this work can be divided into two major components. A genetic algorithm (GA) for fast screening of chemical space and density functional theory (DFT) methods for high-level quantum mechanical energy calculations. The two components are explained in detail below.

### Method—Genetic algorithm

The method deployed for search of chemical space was a graph-based genetic algorithm (GB-GA) and the functionality of crossover and mutation operations on SMILES strings in this study is identical to the one implemented by Jensen. The essential idea of the GA was to modify the Schrock catalyst to create new possible substituents that improve upon the original Schrock catalyst by replacing the HIPT substituents attached to the equatorial amines in the triamidoamine core (Fig. 2). The GA gene is therefore organic molecules with an attachment point indicating where it will be attached to the triamidoamine core as replacement for the HIPT substituent. Here we only consider cases where all three attached substituents are identical. The optimizing objective of the GA is to lower reaction energies between key catalytic steps in the Schrock cycle (see the 'Scoring' section below).

#### *Molecular processing*

The starting population is constructed from randomly selected amines from a 250 K molecule subset of the ZINC database (*Sterling & Irwin, 2015*). The entries in the database

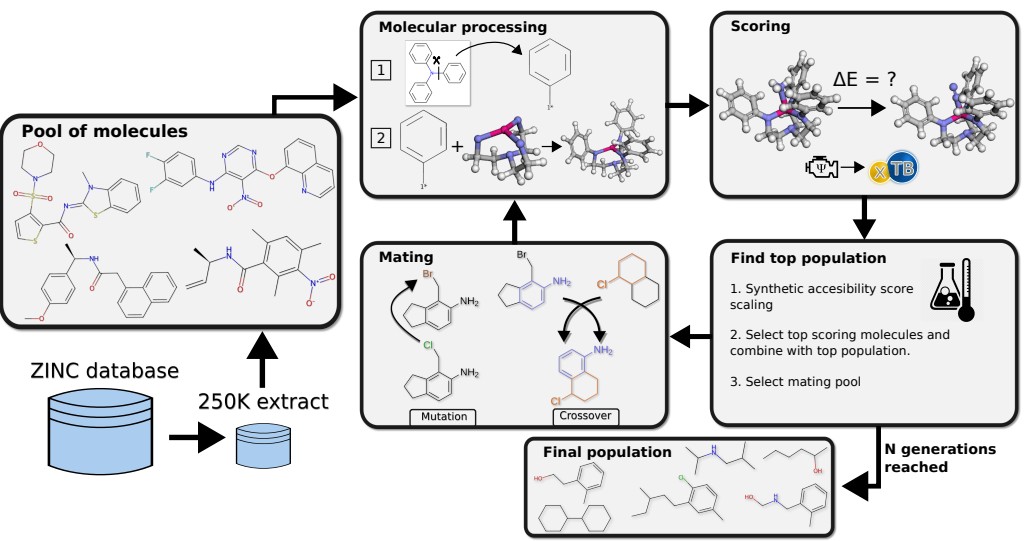

**Figure 2  Workflow of the genetic algorithm.** From a pool of molecules a starting population is created and evolved for N generations, the loop is then terminated and the final population of well scoring candidates is returned.

are all commercially-available compounds, which makes it ideal for virtual screening. From these molecule we extract moieties that were connected to non-ring Nitrogen atoms by single bonds. The single bond that was bound to nitrogen is instead attached to a dummy atom, which is used to indicate the attachment point of the substituent. 50–100 substituents (depending on population size) are then selected at random to form the initial population.

The fitness of each gene is evaluated by attaching three copies to the triamidoamine core and generating a 3D structure using the ETKDG method in RDKit (*Riniker & Landrum, 2015*) (using the embedding parameters found in Table S3) where the coordinates of the core are constrained to match a DFT optimised structure. The core either includes the $NH_3$, $N_2$ or $NH_3$–$N_2$ reacting moieties, depending on which intermediates are needed in the scoring function.

When computing the synthetic accessibility score (see 'Scoring') the dummy atom indicating the attachment point is replaced with a H atom. When performing the mating and mutation operations the dummy atom is replaced by an N atom as the dummy atoms are also used during the mating operations and the mating operations are implemented so that the substituent always contains at least one amine. It is possible that during crossover or mutation, the attachment point is lost, in which case a new attachment point is made from other amines in the molecule.

We found that other primary amines in the substituent (*i.e.,* ones not removed to form an attachment point) tended to form relatively strong bonds to Mo during structure relaxations (see Fig. S6). Such strong interactions are not present at the DFT level and appears to be an artifact of the quantum method used (see scoring). Therefore, such primary amines groups were replaced with a H atom (see Section S3).

### Scoring

The fitness (or score) of a gene is mainly determined by the reaction energy ($\Delta E$) of one of three steps in the Schrock cycle, computed at the GFN$_2$-xTB (*Bannwarth, Ehlert & Grimme, 2019*) level of theory, using the lowest energy structures out of four conformers generated for each intermediate (**Scoring** in Fig. 2).

The xTB optimization of the embedded structures followed a four step procedure, each step using the optimized structure of the previous step as a starting point. This is done to have a stepwise relaxation of the substituent as simply optimizing directly on the initial substituent coordinates from the embedding can lead to faulty optimizations and unwanted intramolecular reactions.

First a GFN-FF (*Spicher & Grimme, 2021*) force-field optimization is performed on the attached substituents, with the core atoms fixed. Then a GFN$_2$-xTB optimization is performed with the same constraints. The third optimization removes the constrains on the core, except on the Mo atom and the attached $N_xH_y$ moiety. During the final optimization, all atoms except the Mo atom and the $N_xH_y$ moiety are constrained. This is done to prevent any detachment of the $N_xH_y$ moieties during optimizations in the GA runs.

From the energies of the optimized structures, reaction energies between the intermediates are obtained according to the reactions stated in Eqs. (1), (2), (3). These represent three sub-reactions from the Schrock cycle, chosen since they are deemed to be the determining factors in the overall reaction, and from here on these are referred to as scoring functions. For simplicity, the scoring functions will be referred to in a reduced form without the molybdenum prefix, for example $NH_3 \rightarrow N_2$.

$$Mo\text{–}NH_3 + N_2 \rightarrow Mo\text{–}N_2 + NH_3 \tag{1}$$

$$Mo\text{–}NH_3^+ + e^- \rightarrow Mo\text{–}NH_3 \tag{2}$$

$$Mo\text{–}NH_3 + N_2 \rightarrow Mo\text{–}NH_3\text{–}N_2. \tag{3}$$

Each reaction energy $\Delta E$ is then multiplied by the score modifier suggested by *Gao & Coley (2020)*, using the synthetic accessibility scoring function developed by *Ertl & Schuffenhauer (2009)*, to help ensure synthetic accessibility (SA).

The current population is merged with the previous population, ranked by score, and the top $N$ unique substituents are selected as the next population, where $N$ is the population size. Finally, the scores of the population are normalized according to Eq. (4).

$$\text{Normalized score}_i = \frac{\text{Score}_i - \text{Max}(\text{Scores})}{\sum_{i=1}^{N} (\text{Score}_i - \text{Max}(\text{Scores}))}. \tag{4}$$

The worst scoring substituent has a normalized score of 0 and the rest a number between 0 and 1, with all scores summing to 1. Substituents are then selected for mating and mutation using roulette selection and these normalized scores. We found that occasionally the bonding in the substituents rearranged and these were discarded by giving them

an artificially high energy score of 9999, ensuring removal from the population. The connectivity is computed based on the overlap charge density from an extended Hückel calculation in RDKit with an overlap threshold of 0.15.

A typical GA search is performed for 50 generations, using a population size of 50, and a mutation rate of 0.5. The usual run time for a GA run with these parameters for 50 generations would be around 5-12 h, depending on the size of the evolved substituents. In general, 8 cpu cores were assigned to each substituent in the population. Thus, for runs with 4 conformers, 2 cores were used for each conformer. We performed 23 GA searches in total. 11 for Eqs. (1), 9 for (2) and 2 for (3). Relevant parameters for all GA runs can be seen in Table S1.

## Method—DFT verification

The substituents in the final populations of the GA searches, obtained using $GFN_2$-xTB, are reevaluated at the DFT level of theory (Fig. 3). For all DFT calculations ORCA 5 (*Neese, 2022*) was used. Following Thimm et al., we used PBE (*Perdew, Burke & Ernzerhof, 1996*)/ZORA-def2-TZVP (*Weigend & Ahlrichs, 2005*; *Pantazis et al., 2008*)/D3BJ (*Grimme et al., 2010*; *Grimme, Ehrlich & Goerigk, 2011*) (SARC-ZORA-TZVP for Mo) for single point evaluations using $GFN_2$-xTB structures as well as for geometry optimisation of select substituents, and B3LYP (*Becke, 1993*; *Becke, 1988*; *Lee, Yang & Parr, 1988*)/ZORA-def2-TZVP/D3BJ for single points using the PBE optimized structures. For PBE calculations the Split-RI-J (*Neese, 2003*) approximation is applied and for B3LYP we used the RIJCOSX (*Neese et al., 2009*; *Izsák & Neese, 2011*) approximation. Relativistic effects are treated with the zeroth order regular approximation (ZORA (*van Lenthe, Baerends & Snijders, 1993*)). See Table S3 for more details. We refer to these levels of theories simply as PBE and B3LYP here after. Thimm et al. used the def2/J auxilliary basis set in their ORCA3 (*Neese, 2012*) calculations, while we used the larger SARC/J basis set recommended for ZORA calculations with ORCA5. Thimm et al. used the COSMO (*Klamt & Schüürmann, 1993*) solvation model, which is no longer available in ORCA5. Instead we used the CPCM (*Barone & Cossi, 1998*) model. Due to computational limitations, $GFN_2$-xTB is used to compute free energy contributions.

Each step in the DFT verification stage is visualized in Fig. 3. The top 10-50 substituents from each of the 23 GA runs were extracted for validation. This led to a pool of 299 substituents. These were re-evaluated with PBE singlepoint calculations in step 2. The energy distribution of these 299 substituents can be seen in Fig. S4. Then, substituents with more than four rotatable bonds were removed, since it is difficult to perform a thorough conformational search on very flexible substituents. Furthermore, we discard substituents with reaction energies for their specific scoring function with $\Delta E > 20$ kcal/mol. This relatively high cutoff is used to minimize the chance of discarding substituents that might have an improved energy score at a higher level of theory. This left us with 141 possible substituent candidates.

Next we perform a more thorough conformational search on the 141 substituents, by re-calculating the scoring function with 100 conformers for each intermediate and optimize with $GFN_2$-xTB (step 3; Fig. 3). Here an additional, fifth optimization, is added in addition

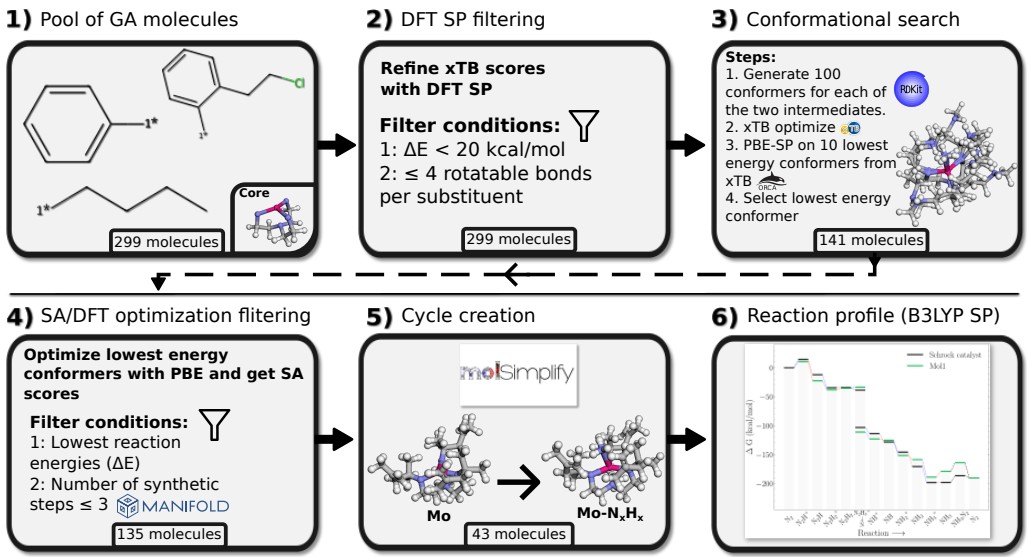

**Figure 3 Workflow for the DFT verification of substituent candidates from genetic algorithm runs.** The bottom label of each tile refers to the remaining pool of molecules at this particular step of the verification process.

to the four optimizations performed during GA runs. The final optimization is performed on the full structure with no constraints on the atoms to allow for a full structure relaxation.

We then perform PBE single point energy evaluations on the 10 lowest GFN$_2$-xTB energy structures and select the lowest PBE energy structure for geometry optimisation at the PBE level of theory. We noticed that xTB optimizations occasionally lead to the detachment of the N$_x$H$_y$ moiety, and such structures tend to have higher energies at the PBE level and are thus discarded at this step (Fig. S7).

GFN$_2$-xTB had been used to obtain low energy conformers in the conformer search, thus the conformers did likely not represent the lowest energy on the DFT surface. As such, the final lowest DFT SP energy conformers were passed to full DFT-PBE optimization (step 4; Fig. 3) in order to obtain the relaxed structures and thereby relaxed reaction energies at the PBE level.

As a last step before final substituent selection, the retrosynthesis tool Manifold (*Anonymous, 2022*) is used to predict the minimum number of synthetic steps required to synthesise each substituent from commercially available building blocks, and discard those with four or more synthetic steps.

After the filtering in step 4 we select the top 15 substituents for scoring functions Eqs. (1) and (2). As there was only 13 substituents left from scoring function Eq. (3) at this point, all of these are selected. For this total of 43 substituents, molSimplify is used to create all 15 catalytic intermediates and these are optimized at the PBE level of theory. This procedure succeeded for 20 of the substituents. See Figs. S9 and S10 for visualization of the 43 substituents. The remaining 23 substituents generally failed due to SCF convergence problems for all or some of the intermediates. In general, we found that the SCF convergence

to be very sensitive to small changes in molecular structure, which can often be fixed by manual intervention. However, as this was not necessary for 20 of the substituents we did not pursue this further. B3LYP singlepoints reaction profiles with GFN$_2$-xTB free energy corrections were obtained for these 20 substituents. From these 20 we then chose 3 for closer examination of the reaction profiles and structures.

## RESULTS

### Reference energies

For modelling the alternating protonation and reduction steps of the Schrock cycle we apply the same procedure as *Thimm et al. (2015)* with lutidinium (Lut) acting as proton donor and decamethylchromocene (Cp$_2$*Cr) acting as electron donor. Calculated energies for both are found in Table S5.

Figure 4A compares the electronic energy profiles of the Schrock catalyst (with the HIPT substituent) obtained in this study to that obtained by *Thimm et al. (2015)*. Figures S2 and S3 contain additional direct comparisons between the reaction energies of all sub reactions. The differences in reaction energies are in the range 0–10 kcal/mol and only the reaction energy of the N$_2$H$^+$ → N$_2$H step deviates by more than 10 kcal/mol. There are three possible reasons for the observed discrepancies: the different solvation model and auxiliary basis set in ORCA3 and ORCA5, and conformation differences (Thimm et al. do not provide coordinates). The overall reaction energies, which corresponds to the following reaction

$$N_2 + 6[Cp_2{}^*Cr] + 6LutH^+ \rightarrow 2NH_3 + 6[Cp_2{}^*Cr] + + 6Lut \tag{5}$$

are nearly identical. Since this involves many charged species, one would expect that this reaction energy is most sensitive to solvation effects. The good agreement thus indicates that differences due to the solvation models are likely to be relatively small (although it should be noted that the electronic energy of CpCr2$^+$ was found to be extremely sensitive to the starting structure). Additional calculations reveal that the effect of the difference in auxiliary basis set have negligible effects on the electronic energy. The main source of the relatively modest difference in the electronic energy profiles shown in Fig. 4A is therefore most likely due to conformational effects.

As mentioned in the previous section, computational limitations prevented us from computing the vibrational free energy corrections at the DFT level of theory. Figure 4B compares the free energy profiles of the Schrock catalyst computed using GFN$_2$-xTB free energy corrections to that obtained by Thimm et al. Comparing Figs. 4A and 4B it is evident that computing the free energy corrections using GFN$_2$-xTB does not introduce bigger discrepancies for the results from Thimm et al. than those due to conformational effects. See Fig. S1 for direct comparison of the vibrational corrections. This matches their findings of the electronic energy differences as the main contributor to the free energy differences.

### DFT verified substituents

As previously mentioned, we are able to obtain complete catalytic cycles at the DFT level of theory for 20 of the GA-generated substituents. For the substituents scored on the last

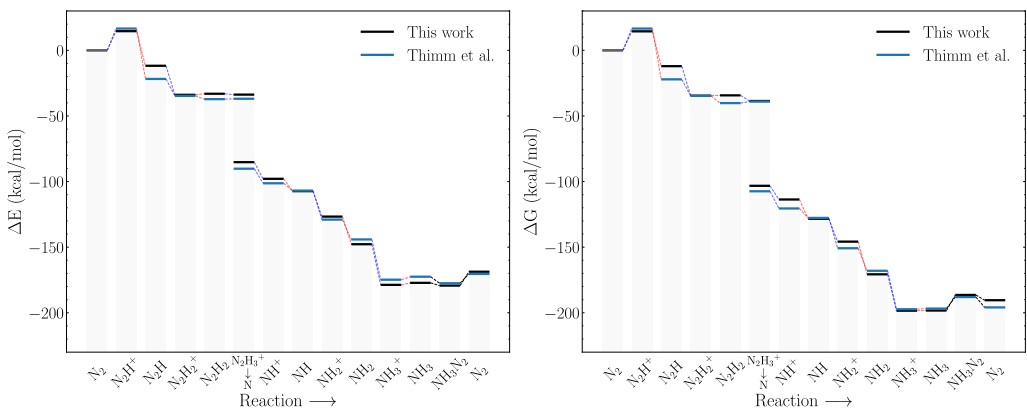

**Figure 4** Reaction profiles for the Schrock catalyst calculated with PBE optimizations and B3LYP singlepoints as compared to *Thimm et al. (2015)*. (A) Electronic energies, (B) Free energies where the energies obtained in this work have been augmented with xTB vibrational corrections instead of DFT. Dotted blue lines indicate proton transfer and red lines indicate electron transfer. The *x*-axis labels refer to the state of the $N_xH_y$ moieties on the molybdenum.

step where $NH_3$ is displaced by $N_2$ five cycles were obtained. Six catalytic cycles were obtained for substituents scored on the last reduction ($NH_3^+ \rightarrow NH_3$), and nine full catalytic cycles for the substituents scored on binding of $N_2$ to form the six-coordinated intermediate (Table 1). The last column in the table indicate the highest or next highest change in reaction energy compared to the reference catalyst. The full reaction profile for each substituent is found in Section S5.2.

The two key questions are whether the xTB electronic energy-based GA search provides favorable reaction free energies at the B3LYP level and, if so, whether the reaction free energies of other steps in the catalytic cycle are affected. For the $NH_3 \rightarrow N_2$ reaction the GA-derived substituents all display more favorable reaction energies than the Schrock catalyst reference (7.86 kcal/mol), while for the $NH_3^+ \rightarrow NH_3$ reduction four of the six substituents have favorable reduction energies. Finally, for the $NH_3 \rightarrow NH_3–N_2$ step, there are three substituents for which the reaction energy is lower than for the HIPT substituent. However, all energy differences are still positive. To see a xTB and DFT comparison of scoring step reaction energies for the 20 substituents see Fig. S8. We select one substituent from each scoring function group for further analysis; these are marked in bold in Table 1. Mol1, Mol8, Mol12 were hand-picked based on their scores and reaction profile.

### Reaction profiles
#### NH₃ and N₂ exchange
The free energy profile of **Mol1** (Table 1) is shown in Fig. 5 together with the PBE optimized 3D structures of the $NH_3$ and $N_2$ intermediates. The energy of the former is indicated by the lower green bar in the column marked $NH_3 \rightarrow NH_3–N_2$ that is connected to the preceding green bar by a red line (indicating reduction). The energy of this structure is 11.44 kcal/mol higher than the $N_2$ structure, whereas the corresponding structure for HIPT substituent is 7.86 kcal/mol lower. The likely reason is that the region around the $NH_3$ is

**Table 1 Overview of substituents with fully converged catalytic cycles at the end of the DFT verification.** The energies in $\Delta G°$ indicate the energy difference of the scoring step used. They were obtained from B3LYP singlepoint calculations on the PBE optimized structures with xTB vibrational corrections. $\Delta\Delta G°_{ref}$ indicates deviations between the highest or next highest free reaction energies of the catalysts compared to the reference Schrock catalyst.

| | | | |
|---|---|---|---|
| **Schrock catalyst:** | $NH_3 \rightarrow N_2$ | 7.86 | |
| | $NH_3^+ \rightarrow NH_3$ | 0.13 | |
| | $NH_3 \rightarrow NH_3{-}N_2$ | 11.87 | |
| SMILES | Scoring/label | $\Delta G°$ | $\Delta\Delta G°_{ref}$ |
| | $NH_3 \rightarrow N_2$ | | |
| [1*]C(C)(C)CCC1CCCCC1 | **Mol1** | −11.44 | 9.88 ($NH_3^+ \rightarrow NH_3$) |
| [1*]C(C)Cc1ccc(Cc2ccccc2)cc1 | Mol2 | −12.32 | 11.66 ($NH_3^+ \rightarrow NH_3$) |
| [1*]C1(CCCCCl)CCCCC1 | Mol3 | −11.84 | 7.82 ($NH_3^+ \rightarrow NH_3$) |
| [1*]C(C)Cc1ccc(CCl)cc1 | Mol4 | −12.45 | 8.76 ($NH_3^+ \rightarrow NH_3$) |
| [1*]C(C)(C)CC(=C)C | Mol5 | −9.80 | 8.80 ($NH_3^+ \rightarrow NH_3$) |
| | $NH_3^+ \rightarrow NH_3$ | | |
| [1*]c1ccccc1N=CC(=O)Cl | Mol6 | −27.28 | 50.10 ($N_2H_2 \rightarrow N_2H_3^+$) |
| [1*]c1cc(C#N)cnc1C#N | Mol7 | −23.39 | 35.21 ($N_2H \rightarrow N_2H_2^+$) |
| [1*]c1c(C#N)ccnc1C(=O)Cl | **Mol8** | −12.43 | 16.56 ($N_2 \rightarrow N_2H^+$) |
| [1*]c1cc(C(=O)CO)cnc1C#N | Mol9 | −2.05 | 18.09 ($N_2 \rightarrow N_2H^+$) |
| [1*]c1cc(CC(=O)O)cnc1C#N | Mol10 | 2.32 | 52.73 ($NH_3 \rightarrow NH_3{-}N_2$) |
| [1*]c1c(C#N)ccnc1C#N | Mol11 | 5.28 | 23.12 ($N_2H \rightarrow N_2H_2^+$) |
| | $NH_3 \rightarrow NH_3{-}N_2$ | | |
| [1*]CCCOc1ccnc2cccnc12 | **Mol12** | 4.57 | 24.98 ($NH_2^+ \rightarrow NH_2$) |
| [1*]CC=Cc1ncnc2ccccc12 | Mol13 | 5.53 | 22.99 ($NH^+ \rightarrow NH$) |
| [1*]CCCc1ncnc2ccccc12 | Mol14 | 10.36 | 84.85 ($N_2H^+ \rightarrow N_2H$) |
| [1*]CCOC(=O)c1ncnc2ccccc12 | Mol15 | 12.81 | 19.86 ($N_2 \rightarrow N_2H^+$) |
| [1*]C=NC(=O)c1cccc(Br)c1 | Mol16 | 15.27 | 25.11 ($N_2H \rightarrow N_2H_2^+$) |
| [1*]CCc1ncnc2ccccc12 | Mol17 | 16.86 | 16.97 ($NH^+ \rightarrow NH$) |
| [1*]CC(=O)c1cc(C(C)=O)ccc1F | Mol18 | 18.48 | 41.51 ($NH^+ \rightarrow NH$) |
| [1*]CCCCOc1ncnc2cccnc12 | Mol19 | 24.48 | 35.47 ($NH^+ \rightarrow NH$) |
| [1*]CC(=O)c1cc(O)c(C(C)=O)cc1O | Mol20 | 35.88 | 33.81 ($N_2H^+ \rightarrow N_2H$) |

**Notes.**

[1*] Denotes the attachment point and all values are in kcal/mol. Substituents marked in bold were selected for further analysis in 'Reaction profiles'. Reaction energies for all 20 catalytic cycles can be found in the supplementary data and the 2D representation of each molecule can be seen in Fig. S10.

sterically crowded compared to the $N_2$ complex. For example, H atoms on the $NH_3$ are as close as 1.95 Å to the H atoms on the nearby methyl groups, whereas the corresponding H-N distance for the $N_2$ complex is 2.45 Å. For comparison the closest H-H distance between $NH_3$ and the HIPT substituent is 2.32 Å, which is also consistent with a less sterically crowded environment around the $NH_3$ and a comparatively lower energy for this intermediate.

Unfortunately, there is not a similar increase in the energy of the $NH_3^+$ intermediate, which results in a 9.8 kcal/mol barrier to reduction (compared to 0.1 kcal/mol for the Schrock catalyst). This barrier (10.6 kcal/mol) is also present in a catalyst where the HIPT substituents are replaced by methyl groups. So, in a sense this barrier is a "canonical"

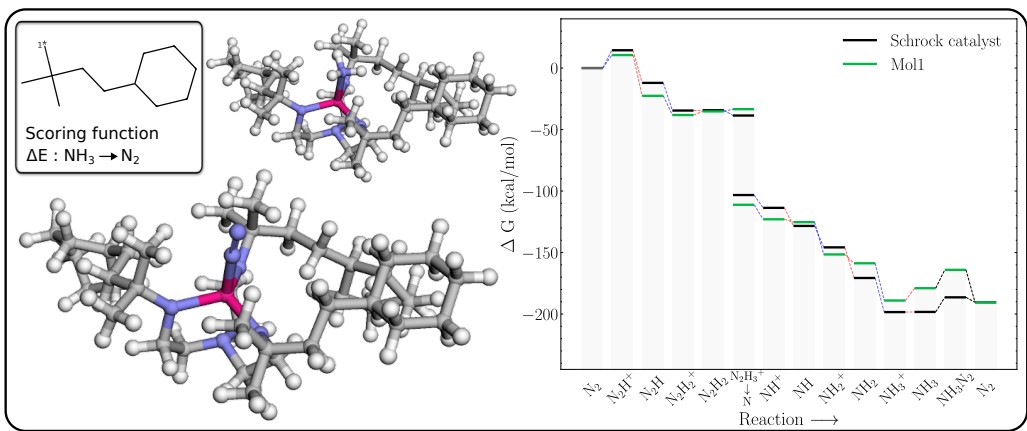

**Figure 5** **Mol1 on the Schrock core (left) and the corresponding energy profile calculated with B3LYP (right) as compared to the energy profile of the Schrock catalyst.** The [1*] on the 2D molecule in the top left corner denotes the attachment point of the molecule. The 3D structure of [Mo–$N_2$] is shown on the lower left and [Mo–$NH_3$] on the top.

barrier presumably due to the decrease in Mo charge upon reduction that results in a weaker Mo-$NH_3$ interaction (the Mo-N distance increases by 0.035 Å).

Note, the role of the Mo catalyst is to lower the energies between each reaction step. The only way for a substituent scored on this last reaction step to achieve this, is to move the $NH_3$ state upwards, as the energy of the $N_2$ state is fixed at the reaction energy for the reaction in Eq. (5). Thus, future GA optimisations on this part of the energy profile should include more intermediates (*e.g.*, Mo-$NH_3^+$) to prevent this barrier to reduction from appearing.

### $NH_3^+$ reduction

The free energy profile of **Mol8** (Table 1) is shown in Fig. 6 together with the PBE optimized 3D structures of the $NH_3^+$ and $NH_3$ intermediates. It is clear that the GA has achieved the objective of making the reduction exergonic, by destabilising the $NH_3^+$ more than $NH_3$.

The structure of the $NH_3^+$ intermediate has short distances between both the Mo and $NH_3$ group, and the Mo and carbonyl oxygens on the substituents which lengthen significantly upon reduction (from ca 2.1 to 2.5 Å), indicating a decrease in the strength of these interactions. We propose that these interactions increases the positive charge on Mo compared to a methyl substituent (with Mullliken charges of 1.92 *vs* 1.66), which increases the electrostatic repulsion with the $NH_3^+$ moiety, leading to a destabilization relative to neutral $NH_3$.

The **Mol8** $NH_3$ intermediate is also slightly destabilized relative to HIPT so the catalysts regeneration is now essentially isogonic (equal in energy).

### $N_2$ binding to form 6-coordinated complex

The free energy profile of **Mol12** (Table 1) is shown in Fig. 7 together with the 3D structures of the $NH_3$ and $NH_3–N_2$ intermediates. The energy of $NH_3–N_2$ structure is 4.6 kcal/mol higher than the $NH_3–N_2$ structure, whereas the corresponding energy difference for the

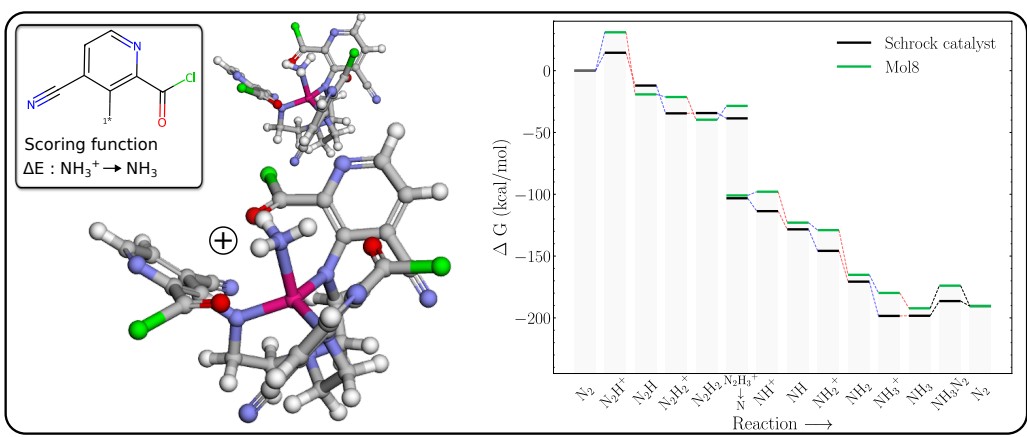

**Figure 6** **Mol8 on the schrock core (left) and the corresponding energy profile calculated with B3LYP (right) as compared to the energy profile of the Schrock catalyst.** The [Mo–NH$_3$]$^+$ is shown at the bottom and [Mo–NH$_3$] at the top.

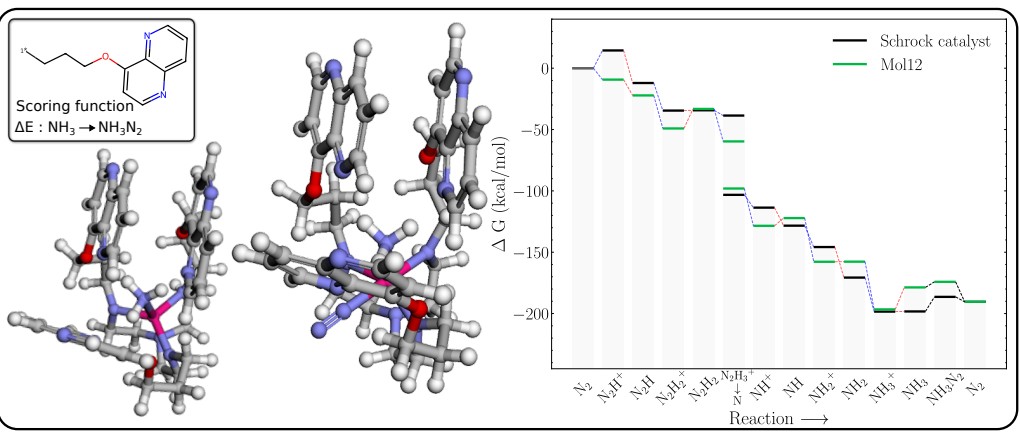

**Figure 7** **Mol12 on the schrock core (left) and the corresponding energy profile calculated with B3LYP (right) as compared to the energy profile of the Schrock catalyst.** The [Mo–NH$_3$] is shown on the left and [Mo–NH$_3$–N$_2$] on the right.

Schrock catalyst is 11.9 kcal/mol. So while the N$_2$ binding is still not exergonic, the GA search manages to significantly lower the energy difference. Furthermore, the N$_2$–Mo distance (2.079 Å) is significantly shorter than for HIPT (3.076 Å, Fig. S12), where the N$_2$ is essentially unbound. As a result of the stronger N$_2$ binding, the axial ligands are roughly in a square planar arrangement with a roughly 180° N$_L$-Mo-N$_L$ angle for the N$_2$ binding site (where N$_L$ is a ligand N). We hypothesize that the cost of increasing this angle is offset by a stronger interaction between the naphthyridine rings. While this can be hard to quantity with individual distances, we note that the surface area of the naphthyridine rings are 1,607 and 1,620 Å$^2$ for NH$_3$–N$_2$ and N$_2$, which supports this assertion.

## CONCLUSION AND OUTLOOK

In conclusion, this work presents a genetic algorithm for in silico catalyst discovery of nitrogen fixation catalysts by searching chemical space for replacements to the HIPT substituent on the Schrock catalyst. From an extract of the ZINC database of 250K molecules, a genetic algorithm based workflow with the $GFN_2$-xTB quantum method was used to discover 299 possible substituent candidates that went through a series of DFT validation steps which resulted in a final pool of 20 substituents for which full PBE-optimized catalytic cycles were obtained. These substituents were observed to lower energies for crucial reaction steps at both the xTB and the B3LYP level of theory. For one scoring function other sub-reactions energies were increased to a minor degree by the introduction of new substituents, for the remaining two scoring functions, other sub-reaction energies were severely increased by the HIPT substituent replacement.

The structures and energy profiles of one promising substituent from each scoring function were examined in greater detail. Each of the three GA evolved substituents were seen to lower the reaction energies for the particular scoring step they were evaluated on. Thus emphasizing the capabilities of the genetic algorithm. The disparity of the substituents from different scoring functions and the varying degree for which they were able to effectively catalyze all sub-reactions highlights the importance of the choice of scoring function. It became evident that scoring on a single reaction step in some cases was a lacking approach as barriers were introduced for other sub-reactions in the catalytic cycle.

Further studies should investigate how an extension of the genetic algorithm scoring functions would impact the quality and disparity of the output substituents. This study has highlighted the importance of considering multiple reaction steps to hinder the introduction of barriers. The scoring function could therefore be extended to include more than two intermediates in order to perform multi-objective optimization of multiple reaction energies. This could either be reaction energies for selected forward and backwards reactions or for separate sub-reactions of the Schrock cycle. Furthermore, future scoring functions could consider the first $N_2$ protonation step and both types of charge transfer (protonation and reduction), in order to prevent the introduction of barriers of sub-reactions not involved in the scoring function. Other things to consider could be the effect of new substituents on the two possible pathways for the $NH_3 \rightarrow N_2$ exchange, or the steric protection of the Mo atom.

### Funding

This work was funded by the Independent Research Fund (DFF; grant number 0217-00326B) and the Novo Nordisk Foundation (grant number NNF20OC0064104). The funders had no role in study design, data collection and analysis, decision to publish, or preparation of the manuscript.

## Grant Disclosures

The following grant information was disclosed by the authors:
Independent Research Fund: 0217- 00326B.
Novo Nordisk Foundation: NNF20OC0064104.

## Competing Interests

Jan H. Jensen is Editor in Chief for PeerJ Physical Chemistry.

## Author Contributions

- Magnus Strandgaard performed the experiments, analyzed the data, performed the computation work, prepared figures and/or tables, authored or reviewed drafts of the article, and approved the final draft.
- Julius Seumer performed the experiments, analyzed the data, performed the computation work, authored or reviewed drafts of the article, and approved the final draft.
- Bardi Benediktsson performed the experiments, analyzed the data, performed the computation work, authored or reviewed drafts of the article, and approved the final draft.
- Arghya Bhowmik conceived and designed the experiments, authored or reviewed drafts of the article, and approved the final draft.
- Tejs Vegge conceived and designed the experiments, authored or reviewed drafts of the article, and approved the final draft.
- Jan H. Jensen conceived and designed the experiments, analyzed the data, authored or reviewed drafts of the article, and approved the final draft.

## Data Availability

The DFT data is available at https://sid.erda.dk/cgi-sid/ls.py?share_id=hp8kfQHWzk.
The code for the genetic algorithm is available at GitHub: https://github.com/jensengroup/.
The data and code are available at Zenodo: Strandgaard, M., & Jensen, J. (2023). Genetic algorithm-based re-optimization of the Schrock catalyst for dinitrogen fixation. Zenodo. https://doi.org/10.5281/zenodo.10074521.

## Supplemental Information

Supplemental information for this article can be found online at http://dx.doi.org/10.7717/peerj-pchem.30#supplemental-information.

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
