# Peer review of "Genetic algorithm-based re-optimization of the Schrock catalyst for dinitrogen fixation"

_PeerJ Physical Chemistry, doi:10.7717/peerj-pchem.30_

## Round 0.1 · original submission · Major Revisions

I believe the referees made some constructive points of criticism that should be addressed in a revised manuscript for publication. Looking forward receiving the revised manuscript at PeerJ.

Reviewer 1 ·

Basic reporting

The manuscript is well written and contextualized. Figures and tables are well made.

Experimental design

The objectives of the manuscript, and the rationale of the employed methodology, are all sound. The manuscript includes sufficient methodological details for reproducible research standards. Supplementary information is provided, as well as the code to run the experiments.

Validity of the findings

The findings, which are mostly methodological in nature (i.e. they introduce and demonstrate an approach) are good and driven to the scientific question in hand.

Additional comments

Some comments that I would like to see assessed by the authors before publication:

1. I wonder why 23 runs were performed per step of choice. It seems like an odd number.
2. Is the charge-density connectivity assignment method better than the alternatives? Im think about both covalent radii and, perhaps more obviously, the XTB-derived connectivity.
3. In L177 i suggest that the use of "generations" is contextualized or replaced by "final generations" or something a bit less misleading, unless I am mistaken.
4. In L272 the quotation marks are wrong.
5. The caption in Table 1 can be clarified. It is not clear to me which and why the steps were chosen for the \Delta\DeltaG column and the reference to supplementary data is vague and can be made specific.
6. Why and on what basis were the 43 ligands selected? Same for 1, 8 and 12 afterwards.
7. Figs. 5-7 seem to have duplicated 3D structures. What is their point?
8. In the outlook it is hinted that multiobjective optimization over several reaction steps would be of interest. What is the advantage of such an approach versus modelling all steps and using a catalytic activity metric, such as the theoretical TOF as derived by Shaik and Kozuch, as a fitness function?

Reviewer 2 ·

Basic reporting

See attached document

Experimental design

See attached document

Validity of the findings

See attached document

Additional comments

See attached document

Annotated reviews are not available for download in order to protect the identity of reviewers who chose to remain anonymous.

---

## Round 0.2 · accepted · Accept

Dear Jan, the primary decision is to accept the paper as is. You are, however free to update the graphics according to the reviewer's suggestions during the proofreading process.

Reviewer 1 ·

Basic reporting

I think the manuscript has been significantly clarified. I do not entirely agree with some of the arbitrary decisions throughout, but overall I think this is fine scientific work that can be published in its current form.

Experimental design

-

Validity of the findings

-

Reviewer 2 ·

Basic reporting

No comment

Experimental design

No comment

Validity of the findings

No comment

Additional comments

While authors have incorporated all the suggestions, I am still a bit unsatisfied with the figures in the paper. Authors have made improvements but basics like the arrow size connecting the blocks (in figure 3) are hidden. I understand the challenges associated with preparing the manuscript in Latex but that does not justify tiny arrows. Remember a reader looks at the figures first before reading and therefore it is extremely to have the graphics appropriate. I leave it on the editor to decide on this aspect.

I also feel authors should provide reference to the NaviCatGS paper from Corminbeauf's group to direct interested readers who might have similar question as me.